# Contribution of the Adenosine 2A Receptor to Behavioral Effects of Tetrahydrocannabinol, Cannabidiol and PECS-101

**DOI:** 10.3390/molecules26175354

**Published:** 2021-09-02

**Authors:** Todd M. Stollenwerk, Samantha Pollock, Cecilia J. Hillard

**Affiliations:** Department of Pharmacology and Toxicology and Neuroscience Research Center, Medical College of Wisconsin, Milwaukee, WI 53226, USA; tstollenwerk@mcw.edu (T.M.S.); smnthpllck@gmail.com (S.P.)

**Keywords:** 4′-fluoro-cannabidiol, cannabinoid tetrad, elevated plus maze, catalepsy, marble bury, HUF-101, equilibrative nucleoside transporter

## Abstract

The cannabis-derived molecules, ∆^9^ tetrahydrocannabinol (THC) and cannabidiol (CBD), are both of considerable therapeutic interest for a variety of purposes, including to reduce pain and anxiety and increase sleep. In addition to their other pharmacological targets, both THC and CBD are competitive inhibitors of the equilibrative nucleoside transporter-1 (ENT-1), a primary inactivation mechanism for adenosine, and thereby increase adenosine signaling. The goal of this study was to examine the role of adenosine A2A receptor activation in the effects of intraperitoneally administered THC alone and in combination with CBD or PECS-101, a 4′-fluorinated derivative of CBD, in the cannabinoid tetrad, elevated plus maze (EPM) and marble bury assays. Comparisons between wild-type (WT) and A2AR knock out (A2AR-KO) mice were made. The cataleptic effects of THC were diminished in A2AR-KO; no other THC behaviors were affected by A2AR deletion. CBD (5 mg/kg) potentiated the cataleptic response to THC (5 mg/kg) in WT but not A2AR-KO. Neither CBD nor THC alone affected EPM behavior; their combination produced a significant increase in open/closed arm time in WT but not A2AR-KO. Both THC and CBD reduced the number of marbles buried in A2AR-KO but not WT mice. Like CBD, PECS-101 potentiated the cataleptic response to THC in WT but not A2AR-KO mice. PECS-101 also reduced exploratory behavior in the EPM in both genotypes. These results support the hypothesis that CBD and PECS-101 can potentiate the cataleptic effects of THC in a manner consistent with increased endogenous adenosine signaling.

## 1. Introduction

∆^9^-Tetrahydrocannabinol (THC) and cannabidiol (CBD) are terpene phenols synthesized by the cannabis plant that can produce therapeutically important effects in humans. For example, evidence is accumulating that THC is an effective analgesic in humans, particularly in the treatment of chronic pain [1], while CBD is currently FDA-approved to treat severe childhood seizures [2]. Human studies also indicate that both THC [3] and CBD [4] can promote sleep and reduce anxiety. Although THC and CBD are structurally very similar and share some therapeutic benefits, there is little overlap in the mammalian proteins with which they interact. THC is a partial agonist with moderate affinity (K_D_s between 40 and 100 nM) at both CB1 and CB2 cannabinoid receptors, and the majority of its pharmacological effects at moderate doses are the result of interactions with these G protein-coupled receptors (GPCRs) [5]. CBD, having greater molecular flexibility, can interact with multiple receptors, including the GPCRs serotonin 1A receptor (5HT1A) and GPR55, as well as several members of the transient receptor potential family of inotropic receptors [6]. The differences in protein targets contribute to the significant differences in adverse effects of THC and CBD. THC, by virtue of its activity as a CB1R agonist, interferes with complex tasks, such as driving, and has dependence liability [7]. On the other hand, the adverse effects of CBD are relatively mild, including somnolence and gastrointestinal disturbances, although incidences of liver toxicity have also been seen [1].

Cellular studies have demonstrated that the equilibrative nucleoside transporter type 1 (ENT-1) is a molecular target that is shared by both THC and CBD. CBD and THC are both competitive inhibitors of ENT-1 nucleoside binding sites with IC_50_ values less than 200 nM [8]. ENT-1 is a major regulator of extracellular and signaling concentrations of adenosine, and ENT-1 inhibitors, including CBD, act as indirect agonists of adenosine receptor signaling [9]. Indeed, multiple studies have demonstrated that some effects of CBD are blocked by adenosine receptor antagonists, including its anti-inflammatory effects [10,11,12].

Interestingly, preclinical studies indicate significant interactions between type 1 cannabinoid receptors (CB1R) and adenosine A2 receptors (A2AR). Both CB1R and A2AR are highly expressed in the striatum, and multiple studies have demonstrated that CB1R and A2AR can form heterodimers [13]. There is functional evidence for interactions between the two systems; for example, the hypolocomotor and rewarding effects of CB1R agonists are diminished by A2AR antagonism [14]. One goal of the studies reported here was to expand this understanding to examine the requirement for A2AR in the effects of THC in the cannabinoid tetrad (locomotor activity, catalepsy, body temperature and spinal pain reflexes) and anxiety assays.

Considerable preclinical and clinical studies have been carried out to describe the interactions between THC and CBD [15]. Published studies suggest that co-treatment with CBD can modulate the effects of THC; however, the mechanisms that underlie these effects are not clear. Important factors include the dose ratios used; the absolute dose of each drug; and the behavioral or physiological response to THC that is measured. Given the data discussed above that some effects of CB1R agonists are modulated by changes in A2AR activity and that CBD can act as an indirect agonist of A2AR, the second goal of these studies was to explore the requirement of A2AR in CBD-induced modulation of THC effects in the cannabinoid tetrad and anxiety assays.

A significant difficulty with the use of CBD as an oral therapeutic is its low and variable oral bioavailability [16]. A series of fluorinated derivatives of CBD have been synthesized in an attempt to increase potency and reduce pharmacokinetic variability [17]. Among these derivatives, 4′-fluoro-CBD (now called PECS-101, formerly called HU-474 and HUF-101) shares many of the pharmacological effects of CBD, including reduced anxiety-like behaviors [17] and reduced responses to painful stimuli [18]. The third goal of these studies was to compare the effects of CBD and PECS-101 as modulators of THC effects in the cannabinoid tetrad and anxiety assays. 

## 2. Results

### 2.1. THC Dose Response Studies

#### 2.1.1. Cannabinoid Tetrad

WT and A2AR-KO mice were injected with vehicle or THC (1, 10 and 100 mg/kg) and then were assayed in the cannabinoid tetrad in the order and with the timing described in the Methods and shown in Appendix A Appendix A.
Locomotor activity: Vehicle-treated, A2AR-KO mice exhibited a greater mean distance traveled in the open field than vehicle-treated, WT mice (Figure 1A). THC produced a dose-dependent reduction in locomotor activity in both WT and A2AR-KO mice. Two-way ANOVA indicated significant effects of both THC (*F*_3,40_ = 10.3, *p* < 0.0001) and genotype (*F*_1,40_ = 7.7, *p* < 0.01) without a significant interaction (*F*_3,40_ = 1.5, *p* = 0.22).Catalepsy: THC produced a dose-dependent increase in the time with front paws on a ring stand, a commonly employed assay for cannabinoid-induced catalepsy (Figure 1B). Two-way ANOVA indicated a significant effect of THC (*F*_3,40_ = 47.4, *p* < 0.0001) and a significant interaction between THC and genotype (*F*_3,40_ = 3.4, *p* < 0.05) without a significant effect of genotype alone (*F*_1,40_ = 2.4, *p* = 0.13). Sidak’s multiple comparison post hoc test revealed that, following treatment with 100 mg/kg THC, A2AR-KO mice exhibited significantly less catalepsy than WT.Body temperature: Rectal temperature was measured as an index of body temperature (Appendix A Appendix A). Two-way ANOVA indicated a significant effect of THC treatment (*F*_3,40_ = 35.0, *p* < 0.0001); genotype did not significantly affect rectal temperature (*F*_1,40_ = 0.9, *p* = 0.33), and the interaction was not significant (*F*_3,40_ = 0.3, *p* = 0.82).Nociceptive reflex: Latency to move the tail in response to a heat stimulus was used to assess the antinociceptive effects of THC in both genotypes (Appendix A Appendix A). Two-way ANOVA indicated a significant effect of THC (*F*_3,40_ = 17.0, *p* < 0.0001); genotype did not significantly affect the tail-flick latency (*F*_1,40_ = 0.9, *p* = 0.35) and the interaction was not significant (*F*_3,40_ = 1.3, *p* = 0.30).

#### 2.1.2. Anxiety Assays

WT and A2AR-KO mice were injected with vehicle or THC (1, 3 and 10 mg/kg). Mice were assessed in the marble bury assay, followed by the elevated plus maze (EPM).
Marble Bury Assay: Two-way ANOVA indicated a significant effect of THC (*F*_3,56_ = 3.2, *p* < 0.05) without a significant effect of genotype (*F*_1,56_ = 0.16, *p* = 0.88) or a significant interaction (*F*_3,56_ = 1.5, *p* = 0.41) (Figure 1C). Dunnett’s *t*-tests indicate that treatment with 3 and 10 mg/kg THC significantly reduces the number of marbles buried compared to vehicle treated in the A2AR-KO mice; there were no significant differences in the WT mice.EPM: Two-way ANOVA indicated no significant effects of either THC (*F*_3,48_ = 2.15, *p* = 0.1) or genotype (*F*_1,48_ = 0.52, *p* = 0.47) and no significant interaction (*F*_3,48_ = 0.74, *p* = 0.53) on the ratio of time spent in the open and closed arms (OAT/CAT) (Figure 1D). There were no significant effects of THC (*F*_3,48_ = 0.30, *p* = 0.73) or genotype (*F*_3,48_ = 0.05, *p* = 0.26) on the total number of arm entries (Appendix A Appendix A).

### 2.2. THC/CBD Combination Studies

WT and A2AR-KO mice were treated with 5 mg/kg of CBD or THC or their combination. All mice received two injections, with vehicle substituting for the drug when required.
Locomotor Activity: The distance traveled in the open field was measured in the eight treatment groups (Figure 2A). Three-way ANOVA indicated significant effects of both THC (*F*_1,55_ = 13.0, *p* < 0.001) and genotype (*F*_1,55_ = 6.3, *p* < 0.05) but not CBD (*F*_1,55_ = 0.025, *p* = 0.87). There was a significant interaction between THC and genotype (*F*_1,55_ = 6.6, *p* < 0.05). Post hoc tests revealed that THC produced a significant reduction in locomotor activity in A2AR-KO mice also treated with CBD.Catalepsy: Three-way ANOVA indicated significant effects of THC (*F*_1,52_ = 20.0, *p* < 0.0001) and CBD (*F*_1,52_ = 5.0, *p* < 0.05) and a trend toward a significant effect of genotype (*F*_1,52_ = 2.4, *p* = 0.13) (Figure 2B). There was a significant interaction between CBD and genotype (*F*_1,52_ = 5.3, *p* < 0.05). Post hoc tests demonstrate that WT mice treated with a combination of THC and CBD exhibited significantly greater time resting on the ring stand than WT mice treated with either drug alone. Neither THC nor CBD or their combination produced catalepsy in A2AR-KO mice. Post hoc tests revealed a significant difference between the THC/CBD cotreatment groups in WT and A2AR-KO mice.Body Temperature: Three-way ANOVA indicates a significant effect of THC (*F*_1,56_ = 4.26, *p* < 0.05) without significant effects of either CBD (*F*_1,56_ = 0.9, *p* = 0.35) or genotype (*F*_1,56_ = 2.6, *p* = 0.11) (Appendix A Appendix A).Nociceptive Reflex: Three-way ANOVA indicates a significant effect of THC (*F*_1,56_ = 19, *p* < 0.001) without significant effects of either CBD (*F*_1,56_ = 0.8, *p* = 0.79) or genotype (*F*_1,56_ = 0.3, *p* = 0.58) (Appendix A Appendix A).Marble Bury Assay. Three-way ANOVA indicates significant effects of both THC (*F*_1,56_ = 10.4, *p* < 0.01) and CBD (*F*_1,56_ = 6.2, *p* < 0.05) but not genotype (*F*_1,56_ = 0.55, *p* = 0.46) (Figure 2C). The interaction between CBD and genotype is significant (*F*_1,56_ = 7.1, *p* < 0.01); post hoc tests demonstrated a significant reduction in the number of marbles buried between the vehicle treated and THC/CBD treated A2AR-KO mice.EPM: Three-way ANOVA indicated that both THC (*F*_1,48_ = 4.9, *p* < 0.05) and CBD (*F*_1,48_ = 5.4, *p* < 0.05) had significant effects on the OAT/CAT ratio, while genotype trended to a significant effect (*F*_1,48_ = 2.9, *p* = 0.09) (Figure 2D). The interaction between genotype and CBD was also significant (*F*_1,48_ = 4.1, *p* < 0.05), while the interaction between CBD and THC trended toward significance (*F*_1,48_ = 2.6, *p* = 0.1). Post hoc tests revealed a significant increase in OAT/CAT in WT mice treated with a combination of THC/CBD compared to those treated with vehicle and those treated with THC alone. Three-way ANOVA of the total arm entries indicated no significant effects of either THC (*F*_1,48_ = 0.5, *p* = 0.49) or CBD (*F*_1,48_ = 1.1, *p* = 0.31) (Appendix A Appendix A). While there was a significant effect of genotype (*F*_1,48_ = 5.0, *p* < 0.05) and a significant interaction between THC and genotype (*F*_1,48_ = 5.9, *p* < 0.05), post hoc tests did not elucidate any significant group differences.

### 2.3. THC/PECS-101 Combination Studies

WT and A2AR-KO mice were treated with 5 mg/kg of PECS-101 or THC or their combination. All mice received two injections, with vehicle substituting for the drug when required.
Locomotor Activity: Three-way ANOVA indicated a significant effect of genotype (*F*_1,53_ = 4.2, *p* < 0.05), while the effects of both THC (*F*_1,53_ = 4.0, *p* = 0.051) and PECS-101 trended to significance (*F*_1,53_ = 3.4, *p* = 0.07) (Figure 3A). There was a significant interaction between THC and PECS-101 (*F*_1,53_ = 6.0, *p* < 0.05) and a trend to an interaction among THC and PECS-101 and genotype (*F*_1,53_ = 3.4, *p* = 0.07).Catalepsy: Three-way ANOVA indicated significant effects of THC (*F*_1,56_ = 12.8, *p* < 0.001) and genotype (*F*_1,56_ = 4.3, *p* < 0.05) but not PECS-101 (*F*_1,56_ = 0.5, *p* = 0.48) (Figure 3B). There was a significant interaction between THC and genotype (*F*_1,56_ = 5.3, *p* < 0.05). Post hoc tests demonstrate that WT mice treated with a combination of THC and PECS-101 exhibited significantly greater time resting on the ring stand than WT mice treated with PECS-101 alone. However, this did not occur in A2AR-KO mice, and there was a significant difference between the WT and A2AR-KO THC/PECS-101 combined treatment groups.Body Temperature: Three-way ANOVA indicates a significant effect of THC (*F*_1,56_ = 4.1, *p* < 0.05) without significant effects of either PECS-101 (*F*_1,56_ = 0.5, *p* = 0.97) or genotype (*F*_1,56_ = 1.9, *p* = 0.17) (Appendix A Appendix A).Nociceptive Reflex: Three-way ANOVA indicates significant effects of THC (*F*_1,56_ = 8.3, *p* < 0.01) and PECS-101 (*F*_1,56_ = 10.8, *p* < 0.01) but not genotype (*F*_1,56_ = 0.3, *p* = 0.87) (Appendix A Appendix A). The examination of the data suggest that THC inhibits the nociceptive reflex in the absence but not in the presence of PECS-101, although the interaction of THC and PECS-101 was not significant (*p* = 0.15).Marble Bury Assay: Three-way ANOVA indicates significant effects of THC (*F*_1,56_ = 10.8, *p* < 0.01) and PECS-101 (*F*_1,56_ = 6.6, *p* < 0.05) but not genotype (*F*_1,56_ = 1.8, *p* = 0.19) (Figure 3C). There was a nearly significant interaction between THC and genotype (*F*_1,56_ = 3.9, *p* = 0.052). Post hoc tests demonstrated that, in the A2AR-KO mice only, THC reduced the number of marbles buried compared to both vehicle and PECS-101 treated mice.EPM: Three-way ANOVA indicates that none of the factors had a significant effect on the OAT/CAT ratio (THC: F_1,48_ = 1.2, *p* = 0.28; PECS-101: *F*_1,48_ = 0.7, *p* = 0.42; and genotype: *F*_1,48_ = 1.0, *p* = 0.34) (Figure 3D). There was, however, a significant interaction between THC and PECS-101 (*F*_1,48_ = 4.8, *p* < 0.05) and a trending interaction among THC, PECS-101 and genotype (*F*_1,48_ = 3.0, *p* = 0.09). Post hoc tests indicated that the combination of THC and PECS-101 significantly reduced the OAT/CAT ratio compared to the effect of PECS-101 alone in the A2AR-KO mice. Surprisingly, PECS-101 had a very significant effect on the total arm entries; the three-way ANOVA results for PECS-101 were *F*_1,48_ = 21, *p* < 0.0001; neither THC (*F*_1,48_ = 2.1, *p* = 0.16) nor genotype (*F*_1,48_ = 0.5, *p* = 0.46) significantly affected the total arm entries (Appendix A Appendix A).

## 3. Discussion

The first goal of the study was to determine whether six behavioral effects commonly seen with CB1R agonist treatment (reduced spontaneous movement, catalepsy, hypothermia, antinociceptive reflex inhibition, marble burying and behavior in the elevated plus maze) were altered in mice with genetic deletion of the A2A subtype of adenosine receptor. We examined a range of THC doses in each of the assays, and the only assay in which a significant interaction between THC treatment and genotype occurred was catalepsy. THC produced significantly less catalepsy in the A2AR-KO mice than WT, suggesting that signaling through the A2AR is required for the full cataleptic effect of THC.

Catalepsy has long been recognized as a cardinal behavioral sign of THC intoxication in rodents [19]. The cataleptic effect of cannabinoid agonists is characterized by immobility when placed in a position that would normally evoke immediate movement. It is not that animals are unable to move, but rather that they are in a trance-like state. It has been suggested that THC-induced catalepsy is responsible for motor vehicle accidents in cannabis-intoxicated individuals [20]. The cataleptic effects of THC and other CB1R agonists are completely dependent on the expression of the CB1 subtype of the cannabinoid receptor [21]. Cell type-specific CB1R deletion strategies [21] and rescue studies in which CB1R are added back to specific neuronal subtypes in otherwise CB1R-null mice [22] both indicate that CB1R expression in D1 dopamine receptor-expressing medium spiny neurons (MSN) of the striatum is sufficient for THC-induced catalepsy. The results of a recent study support this conclusion and further suggest that the CB1R pool responsible is present on mitochondria (mtCB1R) in the axon terminals of D1-expressing striatonigral neurons [23].

Previous studies have demonstrated that CB1R and A2AR can form functionally relevant heterodimers in the striatum [24] and hippocampus [25], and these heterodimers have been suggested to mediate the motor-depressant and addictive effects of the cannabinoid agonists [14]. However, while evidence suggests that the CB1Rs involved in the cataleptic response to THC are in D1R-expressing MSN, A2AR are expressed most abundantly on dendrites of D2 dopamine receptor-expressing MSNs [26,27]. Additionally, A2AR is not found in regions of the striatum that are enriched in D1R-expressing neurons [28]. While studies of the interactions between CB1R and A2AR in striatal slices suggest that A2AR activation positively regulates the synaptic effects of CB1R agonists, the results suggest that the interaction is indirect and not mediated by receptors that are expressed in the same cell [29]. Thus, available evidence suggests that A2AR is critical in the neuronal circuit through which CB1R agonists act to produce catalepsy, but CB1R/A2AR heterodimers are not likely involved. Earlier work from our laboratory demonstrated that THC is an effective inhibitor of the ENT-1, with an IC_50_ value of 170 nM [8]. Therefore, THC-induced catalepsy could be the result of both its CB1R agonist and A2AR indirect agonist effects. Support for this hypothesis comes from data that A2AR agonists are cataleptic per se [30] and can enhance the cataleptic response to haloperidol [31]. This mechanism is consistent with the current finding that the cataleptic effect of THC is reduced in the absence of A2AR.

We have compared the effects of a single combination of THC and CBD to the effects of the same dose of each drug alone in the classic cannabinoid tetrad and on several anxiety-like behaviors. We chose to use a 1:1 dose ratio, as there is evidence that high doses of CBD can affect THC metabolism [19]. As our goal was to explore CBD-mediated enhancement of behavioral responses to THC, we chose to use a threshold dose of THC (5 mg/kg). Indeed, in WT mice, the only consistent effect of 5 mg/kg THC was an increase in tail-flick latency; THC exhibited inconsistent effects in the catalepsy response and did not produce significant effects in the locomotor, temperature, marble bury and EPM assays.

Consistent with previous studies [18,32], 5 mg/kg CBD had no effect on the tetrad behaviors. We also did not see significant effects of CBD at this dose in either the marble bury or EPM assays. Previous findings are mixed in this regard; some earlier studies have shown an anti-anxiety effect of CBD in mice using the EPM [33,34], while others have not [35].

The vehicle-treated A2AR-KO mice travelled significantly greater distances in the open field compared to their WT littermates and, while 5 mg/kg THC did not suppress locomotor activity in the WT, the same dose suppressed locomotor activity in the A2AR-KO mice, particularly in the presence of CBD. These findings suggest that A2AR activity opposes the hypolocomotor effects of THC in WT mice. This finding is at odds with previously published data that striatal A2AR are required for the hypolocomotor effects of cannabinoid agonists [24,29,36].

In WT mice, the combination of THC and CBD produced significantly greater cataleptic behavior than either drug alone. This appears to be a synergistic rather than additive effect, although more extensive dose-response studies are needed to confirm this conclusion. The CBD enhancement of THC-induced catalepsy did not occur in the A2AR-KO mice, suggesting that intact A2AR signaling is required for the effect of CBD, the effect of THC, or the effect of both. Given previous studies that CBD can act as an indirect agonist of adenosine signaling at A2AR in this dose range [8,12], it is possible that CBD enhances THC-mediated catalepsy because it functions as an A2AR activator. This mechanism is analogous to that described above for THC, and we hypothesize that the synergistic effect of CBD on THC-induced catalepsy occurs because CBD can increase adenosine signaling in the circuit.

We also found evidence for a synergistic interaction between THC and CBD to produce an anxiolytic-like response in the EPM that did not occur in the A2AR-KO mice. Previous studies have shown that both the adenosine receptor antagonist, caffeine [37], and the genetic deletion of A2AR [38] are associated with increased anxiety-like behaviors in mice. Importantly, nitrobenzylthioinosine, an ENT-1 inhibitor, produces anxiolytic effects in the EPM when injected into the amygdala [39]. These findings, together with our current results, suggest that the hypothesis that CBD and THC synergize to reduce anxiety and that the ability of one or both of the cannabinoids to inhibit the ENT-1 contributes to this effect under the dose conditions studied.

The third goal of the studies in this project was to compare the effects of 4′-fluoro-CBD (PECS-101) to those of CBD. Earlier studies found that PECS-101 increased open arm time in the EPM assay, reduced immobility in the forced swim assay and enhanced prepulse inhibition at a dose of 3 mg/kg [17]. Higher doses (30 mg/kg and greater) were active in various assays of nociception [18] and exhibited neuroprotective characteristics in rats [40]. In all of these studies, PECS-101 produced effects similar to those of CBD, although was more potent.

We have found similarities between the effects of PECS-101 and CBD in our studies. Like CBD, PECS-101 did not exhibit consistent effects in the tetrad behaviors. Additionally, like CBD, PECS-101 potentiated the cataleptic effects of THC in an A2AR-dependent manner. However, mice treated with PECS-101 exhibited a tendency to bury more marbles regardless of co-treatment with THC or mouse genotype. In addition, mice treated with PECS-101 showed a significant reduction in total arm entries in the EPM in all treatment conditions, suggesting a reduction in exploratory behavior. On the other hand, WT mice treated with PECS-101 trended to increased distance traveled in the open field, suggesting that PECS-101 has a complex effect on locomotor behavior. Analysis of variance indicated that PECS-101 treatment had a significant effect on the tail-flick latency, and examination of the data indicates that it tended to reduce the effect of THC to increase latency. However, post hoc tests did not support a significant difference between THC alone and THC/PECS-101 in either genotype.

Although future studies are required to examine the interaction of PECS-101 with the ENT-1 directly, we hypothesize that it shares the ability of CBD to act as an indirect agonist of the A2AR receptor, particularly in the striatal circuit involved in cataleptic behavior.

There are several limitations of this study. First, it was conducted only in male mice; whether similar interactions occur in female animals is an open and important question. Second, we only examined a single dose combination of THC and CBD/PECS-101. Given the large number of potential targets for CBD and its congeners, it is highly likely that their interactions with the effects of THC will differ at different doses and dose ratios. Finally, because the dose of THC used was low, there is considerable variability in its behavioral effects. In addition, a number of mice in the EPM froze on the open arms and were therefore eliminated from the analysis. This is likely due to the use of a protocol in which the open arms were brightly lit and thus very aversive. This was deliberate in order to potentiate observation of anxiolytic effects but reduced the number of mice per group.

In summary, these findings add to our understanding of the mechanisms of action of THC and potential interactions between CBD and THC. They indicate that, while CB1R agonism is essential to the effects of THC, other mechanisms, including inhibition of adenosine reuptake, could synergize with this mechanism. Importantly, while other studies indicate that CBD and THC could have opposing effects, our data suggest that they produce synergistic effects on catalepsy. Assuming that catalepsy is an undesirable effect of cannabis, these data indicate that combined THC/CBD preparations could be more harmful than either drug alone. In addition, some cannabinoid users combine cannabis and CBD with coffee and other caffeinated beverages to modulate the psychological effects. It is possible that caffeine moderates the THC and/or CBD experience by inhibiting their effects on A2AR-mediated signaling. On the other hand, recent data indicate that activation of A2AR signaling can have beneficial effects in the context of substance use disorders [41], which together with human studies demonstrating that CBD can reduce anxiety and craving in opiate-dependent and abstinent individuals [42], suggests that the ability of CBD to elevate A2AR signaling could be an important therapeutic mechanism.

## 4. Materials and Methods

### 4.1. Animals

All of the animal experimentation reported herein were carried out in accord with ARRIVE guidelines and was approved by the Institutional Animal Care and Use Committee at the Medical College of Wisconsin.

The animal subjects for this study were male adult mice between 8 and 12 weeks of age. All mice were obtained from in-house breeding of 129S-*Adora*2a^tm1Jfc^/J mice; breeders were originally obtained from The Jackson Laboratory, Bar Harbor, ME, USA (Stock Number 010685). The mice were originally developed by Chen and colleagues [43] and are on a mixed background of 129S and C57-Bl6/J. Tissue from the ear pinnae was used as a source of DNA for genotyping; tissue was added to 0.3 mL of 10 mM NaOH containing 1 mM EDTA and heated at 95 °C for 13 min. All mice were genotyped at least once, and genotypes were re-assessed if ambiguous bands were seen. The primers used were common forward primer: GGG CTC CTC GGT GTA CAT; reverse WT primer: CCC ACA GAT CTA GCC TTA; and reverse knock out primer: CAT TTG TCA CGT CCT GCA CGA C. For the WT reaction, samples were held at 94 °C for 2 min; then cycled 35 times (94 °C for 45 s, 56 °C for 30 s, and 72 °C for 2 min) followed by holding at 72 °C for 10 min. For the KO reaction, the cycle temperature and times were 94 °C for 45 s, 60 °C for 1 min and 72 °C for 1 min, followed by 2 min at 72 °C. Male WT and A2AR-KO offspring of het by het breeders were used as experimental subjects in this study. Because of the mixed strain background, the mice exhibited a variety of coat colors, from white to black. We did not use mice with white fur in these studies because it was more difficult for the tracking software to identify the mice, and they tended to exhibit greater sensitivity to the light source in the tail-flick assay.

### 4.2. Drugs

All drugs were delivered by intraperitoneal injection in a volume of 0.1 mL/25 g body weight. Drugs were administered individually, and multiple injections were given as close in time as possible using opposite abdominal sides. The order of drugs and the side given were randomized. Drug emulsions were prepared using an emulphor:ethanol:saline (1:1:18) vehicle as described previously [44]. Briefly, drugs were dissolved in 100% ethanol at a concentration 20 times greater than the final desired concentration. An equal volume of Kolliphor EL (Sigma Chemical, St. Louis, MO, USA, C5135) was added, and the mixture was vortexed well. Sterile saline was added in a dropwise fashion with continuous vortexing. THC and CBD were obtained from the NIDA Drug Supply Program. PECS-101 was obtained from Gary Hiller (Phytecs, Inc., Los Angeles, CA, USA). The studies in which the effect of THC was examined in the behavioral tetrad were carried out using doses of 1, 10 and 100 mg/kg based upon previous dose-response studies [45] and with the goal of dose range-finding. Because 100 mg/kg THC resulted in a complete loss of locomotor activity, we chose to use lower doses (1, 3 and 10 mg/kg) in the anxiety assays as both of these assays require animal movement to be useful. Our choice of 5 mg/kg THC for the combination studies was driven by our goal of investigating additive and synergistic effects of THC and CBD, so a low THC dose was utilized for these studies.

### 4.3. Behavioral Assays

Two sets of mice were used; the cannabinoid tetrad was conducted in one set, and the EPM/MB assays in the second set.

For the tetrad, mice were acclimated to the experimental room for at least 30 min prior to baseline measurements (Appendix A Appendix A). Baseline measurements included body weight, rectal temperature, and pre-treatment latency in the tail-flick assay. Mice were injected at *t* = 0 and allowed to remain undisturbed in their home cage until *t* = 25 min. The mice were placed into the open field at *t* = 25 min and behavior recorded for 15 min, followed immediately by rectal temperature measurement and placement of the mice into the home cage. At *t* = 50 min, the tail-flick assay was conducted, followed at *t* = 60 min by the ring stand (catalepsy) assay.

For the open field assay component of the tetrad, mice were placed into a round plexiglass arena (diameter 19 inches and height 13 inches) that was cleaned with 70% isopropyl alcohol between mice. Behavior was recorded with a ceiling-mounted Sony Handycam (HDR-CX405) and was analyzed by AnyMaze Behavior Tracking Software (Stoelting, Wood Dale, IL, USA). Rectal temperature measurements were determined at baseline and after drug treatment; a thermistor probe (Physitemp RET-3 probe and BAT-12 thermometer, Clifton, NJ, USA) was lubricated and inserted to a depth of 25 mm and held in place until a stable reading was obtained. For the tail-flick reflex, latency to move the tail away from a heat-generating light (IITC Tail Flick Analgesia Meter Series 8, Woodland Hills, CA, USA) was recorded, and a cut-off time of 10 s was used to prevent injury to the tail. For the ring stand assay, mice were placed with their front paws on a metal ring 4.5 cm above the bench top, and the time to remove both paws was recorded with a stopwatch.

For the anxiety assays (which were carried out in a separate set of mice), mice were habituated to the testing room for at least 30 min, followed by injections and return to the home cage for 30 min (Appendix A Appendix A). The marble bury assay was completed first. Mice were placed individually in clean cages containing 5 cm of bedding that was smoothed and slightly compacted; twenty-four 1.5 cm blue marbles were arranged in a 4 × 6 array on top of the bedding. After 30 min with full room lighting, mice were removed, and the number of marbles covered to a depth of at least 2/3 with bedding was recorded. After 25 min in the home cage, the mice were placed in the center space of an elevated plus maze and allowed to explore for 5 min. The EPM apparatus consisted of two open arms (30 cm long × 5 cm wide) and two enclosed arms (30 cm × 5 cm × 15 cm walls) elevated 40 cm from the floor. Room lighting was turned off, and the open arms of the EPM were lighted. Behavior on the EPM was recorded using the ceiling-mounted camera described above and analyzed using AnyMaze Behavior Tracking Software.

### 4.4. Statistical Analyses

Statistical analyses were carried out using Prism version 9 (GraphPad). All data sets were analyzed using ANOVA followed by post hoc tests if appropriate. For the THC dose response studies, 2-way ANOVA was used with drug dose and genotype as the factors; for the drug combination studies, 3-way ANOVA was used with THC, CBD (or PECS-101) and genotype as the factors. Sidak’s Multiple Comparison’s post hoc tests were used when significant interaction terms occurred. In one study (the effects of THC on marble burying response), Dunnett’s *t*-tests were used to compare the effects of THC doses to the vehicle group. Statistical information for the interaction terms in the 3-way ANOVAs is only provided if significant. In all studies, the original number of replicates was 8; however, outlier analysis (ROUT method with Q = 1.0) was applied to each set of replicates which occasionally resulted in a reduction in n. For the EPM studies, there were multiple instances of mice that entered and remained immobile in the open arm. These mice were removed from the EPM analysis, although their marble bury response was still analyzed.

## Figures and Tables

**Figure 1 molecules-26-05354-f001:**
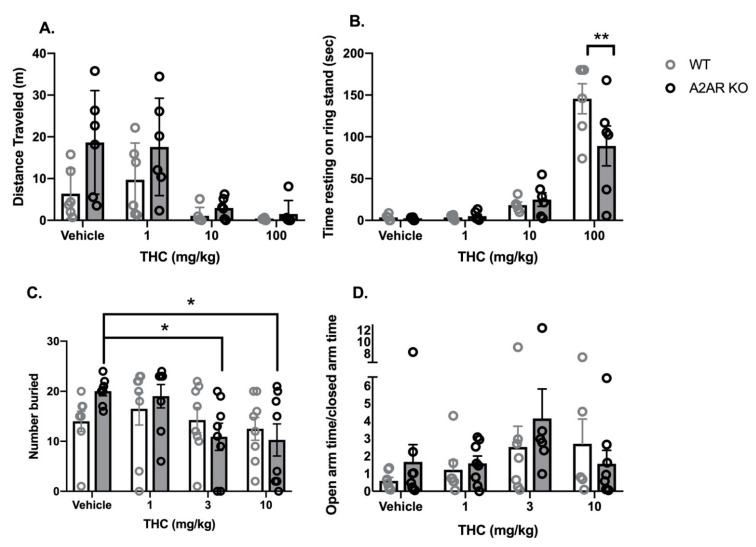
Comparison of some behavioral effects of THC in wild type (WT, open bars and gray symbols) and A2AR null (A2AR-KO, closed bars and black symbols) mice. (**A**) Effects of THC on locomotor activity in an open field. Mice were placed into a circular field for 15 min, and the distance moved was determined. (**B**) Effects of THC on cataleptic behavior in the ring stand assay. Sidak’s multiple comparison test was used to compare all groups to each other. (**C**) Effects of THC on the number of marbles buried. Dunnett’s multiple comparison test was used to compare each drug group to the vehicle control. (**D**) Effects of THC on the ratio of time spent in the open and closed arms of the EPM. A total of 8 mice were removed from this analysis because they entered the open arm and became immobile (1 mouse each from the WT/vehicle, WT/1 mg/kg, WT/3 mg/kg groups; 3 from the WT/10 mg/kg group and 2 from the KO/3 mg/kg group). Bars represent the mean, and vertical lines are the standard error of the mean. * *p* < 0.05 and ** *p* < 0.01.

**Figure 2 molecules-26-05354-f002:**
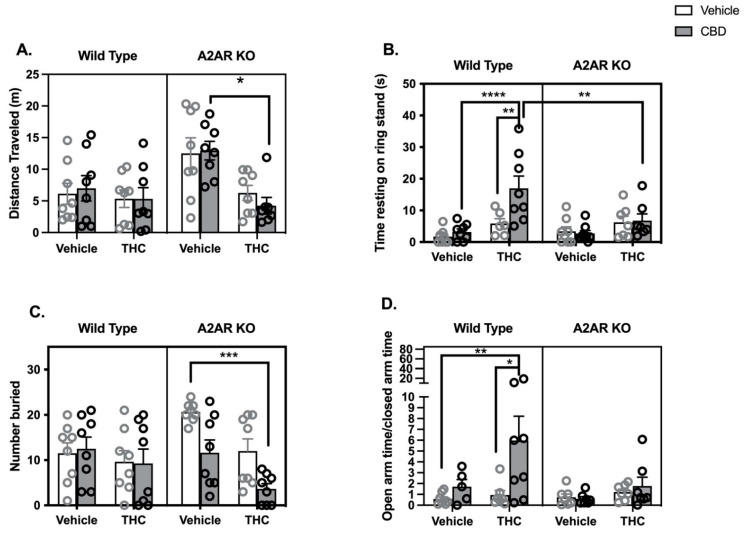
Behavioral effects of THC (5 mg/kg), CBD (5 mg/kg) and their combination in wild type (WT, open bars and gray symbols) and A2AR null (A2AR-KO, closed bars and black symbols) mice. (**A**) Drug effects on distance moved in 15 min while in a cylindrical open field arena. One outlier was identified in the CBD/THC/A2AR-KO group. (**B**) Drug effects on cataleptic behavior in the ring stand assay. Outlier analysis indicated 4 outliers in this data set (2 in the WT/VEH/THC group, 1 in the KO/VEH/THC group and 1 in the KO/THC/CBD group). (**C**) Drug effects on the number of marbles buried. (**D**) Drug effects on the ratio of time spent in the open and closed arms of the EPM. A total of 8 mice were removed from the analysis; 3 mice became immobile on the open arm (2 in the WT/veh/CBD group and 1 in the KO/THC/CBD group); and 5 additional mice were identified as statistical outliers (1 each in the WT/veh/CBD, WT/veh/THC, KO/veh/veh, KO/veh/THC and KO/veh/CBD groups). Bars represent the mean, and vertical lines are the standard error of the mean. * *p* < 0.05, ** *p* < 0.01, *** *p* < 0.001, **** *p* < 0.0001.

**Figure 3 molecules-26-05354-f003:**
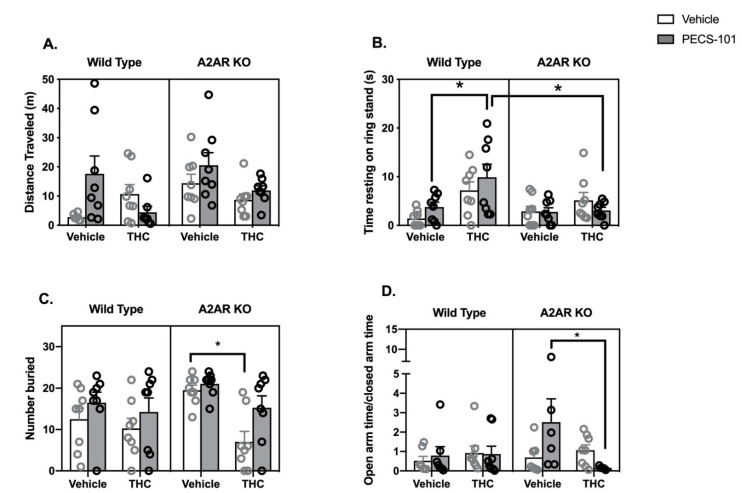
Behavioral effects of THC (5 mg/kg), PECS-101 (5 mg/kg) and their combination in wild type (WT, open bars and gray symbols) and A2AR null (A2AR-KO, closed bars and black symbols) mice. (**A**) Drug effects on distance moved in 15 min while in a cylindrical open field arena. Three outliers were identified (2 in the WT/veh/veh group and 1 in the WT/THC/PECS-101 group). (**B**) Drug effects on cataleptic behavior in the ring stand assay. (**C**) Drug effects on the number of marbles buried. (**D**) Drug effects on the ratio of time spent in the open and closed arms of the EPM. A total of 8 mice were eliminated from this data set; three mice froze on the open arm of the maze (2 in the KO/veh/PECS-101 group and 1 in the KO/THC/PECS-101 group) and five mice were statistical outliers (one each in the WT/veh/veh, WT/veh/PECS-101 and KO/veh/veh groups; and 2 in the KO/THC/PECS-101 group). Bars represent the mean, and vertical lines are the standard error of the mean. * *p* < 0.05.

## Data Availability

Authors will provide raw data upon request.

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
