# Peer review of "Contribution of the Adenosine 2A Receptor to Behavioral Effects of Tetrahydrocannabinol, Cannabidiol and PECS-101"

_molecules, 2021, doi:10.3390/molecules26175354_

Round 1
Reviewer 1 Report
The manuscript titled: „Contribution of the adenosine 2A receptor to behavioral effects of tetrahydrocannabinol, cannabidiol and PECS-101” is original paper in which Authors presents results of expriments performed on wild type mice and A2A knock out mice. Authors demonstrated that A2A receptors are involved in some effects of cannabis and that THC and cannabidiol produce synrgistic effect on catalepsy and that some interaction between THC and cannabidiol may occur.
In my opinion the manuscript is well prepared, results and conclusions are properly defined.
I have some minor recommendations:
- In Introduction Authors should provide the similarities and differences in pharmacological activity of THC, cannabidiol and PCES-101; and point the potential clinical applications;
- There is no data about agreement of Ethical Commitee for behavioral experiments in experimental animals;
- Lack of data about time intervals between drug application and experimental procedures;
After corrections that manuscript can be accepted for publication.
Author Response
In Introduction Authors should provide the similarities and differences in pharmacological activity of THC, cannabidiol and PCES-101; and point the potential clinical applications;
Thank you for the suggestion, we have expanded the information provided regarding the pharmacological activities of the three compounds and have added some potential clinical applications for THC and CBD.
- There is no data about agreement of Ethical Commitee for behavioral experiments in experimental animals
This information was provided in a different section but has been added to the Methods section as well.
- Lack of data about time intervals between drug application and experimental procedures;
We have added timelines to the supplemental information to better indicate the timing between treatment and behavioral assays.
Reviewer 2 Report
This is a well conducted and well reported study focused on a topic of interest.
Author Response
Thank you for the positive response!
Reviewer 3 Report
Most of the binding reports on THC, the active constituent in Cannabis, are on cannabinoid receptors CB1 and CB2. In the submitted paper Stollenwerk et al discuss the effects of THC, CBD and a CBD derivative (PECS-101) due to binding to the adenosine 2A receptor (A2AR). This is of considerable significance in particular since CBD does not bind to CB1 or CB2. Indeed the mechanism of CBD action is not clear at present.
The authors compare numerous activities in wild type and A2AR knock out mice: the tetrad of activities typical for cannabinoids as well as the marble bury assay. The experimental part is well done and the presentations and figures are easy to follow.
The results indicate that the well known THC catalepsy is due not only to CB1 binding but also to its A2AR agonism. I was also impressed by the conclusion that CBD enhances THC catalepsy due to action on A2AR.
The references are relevant. I suggest that the authors should refer to more recent reviews. There are several such reviews published over the last 2 years.
Author Response
Thank you for the positive comments.
We have added more recent reviews of the pharmacology of THC and CBD and the interactions of cannabinoids with adenosine signaling.
Reviewer 4 Report
In the reviewed study, Stollenwerk and colleagues examine the role of adenosine A2A receptor activation in the behavioral effects of THC alone and combination with CBD or PECS-101 using wild type and A2AR knock-out mice. The authors indicate that CBD and PECS-101 can potentiate the cataleptic effects of THC in a manner consistent with increased endogenous adenosine signaling. The presented study is interesting but requires some improvements before potential publication.
Abstract:
1. I suggest expanding the background by adding information on the origin and use of these compounds in humans, and why these compounds are of clinical interest.
2. Please expand the abbreviation THC and CBD in the first sentence of the abstract, as well as the method of administration of THC and CBD.
Introduction:
1. I propose to supplement the Introduction with information on the clinical effects of the use of these (THC and CBD) substances in humans (including psychoactive activity), their negative effects, and therapeutic potential, thus emphasizing the importance of the research undertaken.
2. In the Introduction, I suggest adding brief information about the possibility of the formation of CB1R-A2AR heteromeric complexes and their participation in the pharmacological effects, which is later developed in the discussion (see more in reviews, e.g.: https://www.mdpi.com/2073-4409/9/6/1372).
3. Why was the PECS-101 compound chosen for the research?
4. Line 31: Please add an abbreviation for cannabidiol.
Results:
1. In the description of the results, instead of "n.s." please provide specific values for "p".
2. Line 100: EPM abbreviation appears for the first time in the main text - please expand it in this part.
3. Line: 110: please remove duplicate "(".
4. Figures: Please add in the description of the figures that the bars represent the mean ± SEM or SD.
5. Figure 2: In Part A, please change WILD Type to Wild Type.
6. Line 197: Please separate "PECS-101interaction".
7. Line 205: Please standardize the type of parentheses.
Discussion:
1. I suggest highlighting the clinical implications that may result from this study.
2. Line 250: Please change A2A adenosine receptors to A2ARs.
3. Sentence: PECS-101 is the 4'-fluorinated derivative of CBD (formerly referred to as HUF-101) is a repetition of information from the Introduction section.
Materials and methods:
1. Why in some behavioral tests used doses of THC in the range of 1, 10, and 100 mg/kg, and in others 1, 3, and 10 mg/kg?
2. On what basis were the doses of the individual substances selected for behavioral studies?
3. Line 392: Please add "min" in descriptions for "t".
4. Line 406: It is unclear whether subsequent behavioral tests were performed on one set of animals or a different set of mice.
5. Please complete the information on the software used for the statistical analysis and the post-hoc tests used.
References:
1. References no. 20 and no. 31 are the same article. Please remove one item from the list and update the numbering in the main text.
Author Response
Thank you for the thorough and helpful review.
Abstract:
- I suggest expanding the background by adding information on the origin and use of these compounds in humans, and why these compounds are of clinical interest.
We added this sentence:
The cannabis-derived molecules, ∆9 tetrahydrocannabinol (THC) and cannabidiol (CBD) are both of considerable therapeutic interest for a variety of purposes, including to reduce pain and anxiety and increase sleep.
- Please expand the abbreviation THC and CBD in the first sentence of the abstract, as well as the method of administration of THC and CBD.
DONE
Introduction:
- I propose to supplement the Introduction with information on the clinical effects of the use of these (THC and CBD) substances in humans (including psychoactive activity), their negative effects, and therapeutic potential, thus emphasizing the importance of the research undertaken.
The following sentences have been added:
∆9-Tetrahydrocannabinol (THC) and cannabidiol (CBD) are terpene phenols synthesized by the cannabis plant that can produce therapeutically important effects in humans. For example, evidence is accumulating that THC is an effective analgesic in humans, particularly in the treatment of chronic pain [1] while CBD is currently FDA approved to treat severe childhood seizures [2]. Human studies also indicate that both THC [3] and CBD [4] can promote sleep and reduce anxiety.
AND
The differences in protein targets contributes to the very significant differences in adverse effects of THC and CBD. THC, by virtue of its activity as a CB1R agonist, interferes with complex tasks, such as driving, and has dependence liability [7]. On the other hand, the adverse effects of CBD are relatively mild, including somnolence and gastrointestinal disturbances, although incidences of liver toxicity have also been seen [1].
- In the Introduction, I suggest adding brief information about the possibility of the formation of CB1R-A2AR heteromeric complexes and their participation in the pharmacological effects, which is later developed in the discussion (see more in reviews, e.g.: https://www.mdpi.com/2073-4409/9/6/1372).
The following has been added:
Both CB1R and A2AR are highly expressed in the striatum, and multiple studies have demonstrated that CB1R and A2AR can form heterodimers [13].
- Why was the PECS-101 compound chosen for the research?
We were funded to do this study in part by Phytecs, Inc. The company is interested in determining similarities and differences between CBD and PECS-101 because PECS-101 is more likely than CBD to have a path forward as a pharmaceutical.
- Line 31: Please add an abbreviation for cannabidiol.
DONE
Results:
- In the description of the results, instead of "n.s." please provide specific values for "p".
DONE
- Line 100: EPM abbreviation appears for the first time in the main text - please expand it in this part.
DONE
- Line: 110: please remove duplicate "(".
DONE
- Figures: Please add in the description of the figures that the bars represent the mean ± SEM or SD.
DONE
- Figure 2: In Part A, please change WILD Type to Wild Type.
DONE
- Line 197: Please separate "PECS-101interaction".
DONE
- Line 205: Please standardize the type of parentheses.
DONE
Discussion:
- I suggest highlighting the clinical implications that may result from this study.
The following has been added:
On the other hand, recent data indicate that activation of A2AR signaling can have beneficial effects in the context of substance use disorders [41], which together with human studies demonstrating that CBD can reduce anxiety and craving in opiate dependent and abstinent individuals [42] suggests that the ability of CBD to elevate A2AR signaling could be an important therapeutic mechanism.
- Line 250: Please change A2A adenosine receptors to A2ARs.
DONE
- Sentence: PECS-101 is the 4'-fluorinated derivative of CBD (formerly referred to as HUF-101) is a repetition of information from the Introduction section.
Reworded so not to be repetitive
Materials and methods:
- Why in some behavioral tests used doses of THC in the range of 1, 10, and 100 mg/kg, and in others 1, 3, and 10 mg/kg?
The following was added:
The studies in which the effect of THC was examined in the behavioral tetrad were carried out using doses of 1, 10 and 100 mg/kg based upon previous dose-response studies [45] and with the goal of dose range-finding. Because 100 mg/kg THC resulted in a complete loss of locomotor activity, we chose to use lower doses (1, 3 and 10 mg/kg) in the anxiety assays as both of these assays require animal movement to be useful. Our choice of 5 mg/kg THC for the combination studies was driven by our goal of investigating additive and synergistic effects of THC and CBD so a low THC dose was utilized for these studies.
- On what basis were the doses of the individual substances selected for behavioral studies?
We chose to keep the dose of CBD low because of potential metabolic interactions with THC and a one-to-one dose ratio largely because that is what is used in the nabiximol preparation, Sativex. The dose of PECS-101 was kept the same to allow for comparison with CBD.
- Line 392: Please add "min" in descriptions for "t".
DONE
- Line 406: It is unclear whether subsequent behavioral tests were performed on one set of animals or a different set of mice. Added to the beginning of the section
We have clarified that the tetrad was done in one set of mice and the anxiety assays were done in another.
- Please complete the information on the software used for the statistical analysis and the post-hoc tests used.
DONE
References:
- References no. 20 and no. 31 are the same article. Please remove one item from the list and update the numbering in the main text
DONE
Round 2
Reviewer 4 Report
The authors have adequately addressed my initial concerns about the manuscript. Thank you for your work to clarify these points.